# Joint Factor Performance Validity?—Network and Factor Structure of Performance Validity Measures in the Clinical Evaluation of Adult ADHD

**DOI:** 10.3390/bs15081108

**Published:** 2025-08-15

**Authors:** Emily Raasch, Anselm B. M. Fuermaier, Johanna Kneidinger, Björn Albrecht, Hanna Christiansen

**Affiliations:** 1Department of Clinical and Developmental Neuropsychology, University of Groningen, 9712 TS Groningen, The Netherlands; raasche@staff.uni-marburg.de; 2Department of Clinical Child and Adolescent Psychology and Psychotherapy, Philipps University Marburg, 35032 Marburg, Germany; johanna.kneidinger@uni-marburg.de (J.K.); bjoern.albrecht@uni-marburg.de (B.A.); christih@staff.uni-marburg.de (H.C.)

**Keywords:** adult ADHD, neuropsychological assessment, performance validity, symptom validity

## Abstract

Performance validity tests (PVTs) and symptom validity tests (SVTs) are central to evaluating neuropsychological test results in clinical adult ADHD assessments. Although their relationships have been widely examined, the constructs these measures assess remain poorly understood in applied contexts. This study investigates the conceptual similarities and distinctions of performance validity measures, i.e., the Groningen Effort Test (GET), the Medical Symptom Validity Test (MSVT), and the Amsterdam Short-Term Memory (ASTM) test, within a comprehensive diagnostic battery for adult ADHD. The diagnostic battery included symptom self-reports and a continuous performance test (CPT). Network and factor analyses investigated these relationships. A three-factor structure was hypothesized, consisting of (1) performance validity measures, (2) continuous performance measures, and (3) symptom reports (including embedded SVTs). Data from a large clinical referral sample (*N* = 461) of adults with suspected ADHD were analyzed to explore these constructs. Network analysis revealed that the PVTs did not form a cohesive network with CPT measures. Symptom reports, including embedded SVTs, formed their own cluster, separate from performance-based attention measures. Factor analysis rejected a unified construct of performance validity measures. Regression analysis showed that cognitive deficits, education level, and impulsivity predicted ASTM test performance, whilst the MSVT and GET did not. These findings suggest that PVTs should be interpreted in the context of ADHD assessment, particularly in high-stakes forensic evaluations, where the accuracy of performance evaluation is critical. Future research should explore multidimensional models of performance validity, addressing domain-specific underperformance and individual variability in ADHD evaluations.

## 1. Introduction

Attention-deficit/hyperactivity disorder (ADHD) is a frequent neurodevelopmental disorder, marked by the three core symptoms of inattention, hyperactivity, and impulsivity ([3]). ADHD is increasingly understood as a potential lifespan disorder that begins in childhood. Research indicates that 50% to 65% of individuals diagnosed in childhood continue to experience symptoms in adulthood ([30]; [66]). Unfortunately, many cases go undiagnosed until early or mid-adulthood, making it harder to diagnose early onset when childhood ADHD was not established ([90]). Therefore, diagnosing adult ADHD requires a thorough evaluation of both current and early-life symptoms, along with an assessment of impairment ([74]; [91]). While the core symptoms of ADHD may persist into adulthood, they often manifest in more subtle and diverse ways. These include ongoing challenges with sustained attention, disorganization, inattention to detail, distractibility, and forgetfulness, with motor hyperactivity typically decreasing over time ([107]). If left untreated, these described difficulties can lead to significant impairments across multiple areas of daily functioning, such as academic aspirations ([5]), occupational issues ([1]), and difficulties in social–emotional contexts ([56]). Furthermore, comorbidity rates of up to 80% are common, often involving substance use disorders, depression, anxiety, and personality disorders ([1]; [14]). Given this clinical complexity, it is essential to accurately assess the consistency of information provided by, or about, first-time adult ADHD clients.

To address this need, empirically informed guidelines emphasize a combination of approaches and instruments to ensure a reliable diagnosis. These are designed to enhance accuracy and reduce the risk of false positives and negatives. Such approaches include (structured) diagnostic interviews, following DSM criteria, and standardized self-report rating scales ([90]). Additionally, adult ADHD is associated with cognitive difficulties that affect various aspects of attention and executive control, which, while common in ADHD, are not specific to the disorder ([43]; [74]). Neuropsychological tests, particularly the continuous performance test (CPT), are, however, often used to assess sustained attention in individuals suspected of ADHD ([2]; [4]). Integrating the CPT with other diagnostic tools enhances the diagnostic process and supports ADHD treatment planning ([34]; [44]).

However, these clinical tools are only useful in diagnosis and treatment if they accurately reflect true symptomatology and cognitive abilities. Arising concerns over the credibility of self-reported symptoms and neuropsychological test performance in adult ADHD evaluations have led to the implementation of performance validity tests (PVTs) and symptom validity tests (SVT; [48]; [71]; [100]). Performance validity refers to the assessment of an individual’s actual task performance, which reflects their true cognitive abilities. In contrast, symptom validity refers to the extent to which an individual’s self-and observer-reported symptoms accurately reflect their actual experiences ([60]; [102]). Invalid results of PVTs and/or SVTs may suggest that a patient’s test profile is uninterpretable, but it does not provide insight into the underlying cause of this failure ([68]). This is especially problematic in adult ADHD, where individuals may report ADHD-like symptoms to gain perceived benefits, such as non-medical use of stimulant medication, excuse-making behaviors ([45]; [87]; [96]), or advantages in academic or workplace settings ([1]). Thus, distorted symptom reports and test performance can indeed compromise diagnostic assessments, misguide treatment plans, and reduce adherence ([49]; [81]).

Although SVT research in adult ADHD assessments is less extensive, it is crucial. This is evident, as studies show that common self-report scales often fail to distinguish genuine from simulated ADHD ([100]). Recently, [75] ([75]) reported that 22% of adults undergoing ADHD evaluations failed symptom validity measures. This highlights the urgent need for effective detection methods ([37]). A promising SVT is the Infrequency Index (CII) that is embedded in the Conners’ Adult ADHD Rating Scale ([18]; [94]).

In parallel, PVT research has shown that these tests, alongside SVTs, assess different but complementary constructs. Hence, both should be measured simultaneously in clinical evaluations ([48]; [75]). Drawing from recent large-scale research, estimations of base rates of cognitive underperformance in adult ADHD assessments range from 9 to up to 27% following the currently claimed criterion for underperformance of ≥2 failed stand-alone PVTs ([48]; [67]; [75]; [76]). Stand-alone PVTs offer the highest classification accuracy for detecting noncredible performance in neuropsychological assessments, as they are specifically designed for this purpose ([49]; [103]). Examples of such tests include verbal memory tests (e.g., Word Memory Test; [39]), non-verbal memory tests (e.g., Test of Memory Malingering; [99]), and working memory tests (e.g., Rey Dot Counting Test; [11]). Still, the majority of stand-alone PVTs are not specifically developed for the assessment of adults with ADHD. An exception includes the Groningen Effort Test (GET), serving especially as an attention-based PVT ([35]). In general, it is recommended to select a range of PVTs that address different cognitive domains, as individuals may underperform in specific areas but not others ([10]). To ensure meaningful assessment, the tests within the battery should not be highly correlated, though some degree of overlap is expected ([82]). While a PVT may appear subjectively difficult, they are designed to be easily achievable, minimizing external influences and ensuring optimal test performance ([63]). The underlying concept is that most participants should perform at or near their best level, with those who underperform likely to fail the PVT.

With the increasing use of PVTs and SVTs in clinical assessments, it is essential to clarify their conceptual similarity, particularly in adults with suspected ADHD. Although PVTs vary in design, all aim to identify underperformance in diagnostic evaluations. Therefore, understanding and defining the conceptual structure of PVTs, alongside SVTs, within a comprehensive neuropsychological evaluation is crucial. Analyzing their structure determines whether they measure distinct factors, thereby improving the accuracy of first-time ADHD assessments and providing valid diagnostic information.

While many studies have used prediction and discrimination methods to examine the relationship between PVTs and SVTs ([92]; [105]), fewer have explored the underlying constructs of these tests ([7]; [73]; [86]; [102]). For instance, [102] ([102]) applied a confirmatory factor analysis (CFA) to assess the factor structure of PVTs and SVTs. Their model revealed three factors, consisting of cognitive performance, performance validity, and symptom self-report, with SVTs loading onto the latter. Recently, [7] ([7]) also concluded that PVTs and SVTs represent distinct constructs. Overall, both [102] ([102]) and [7] ([7]) recommended further explorations of validity measures alongside standard diagnostic practices. But neither specifically examined adult ADHD. Given the clinical presentation of adult ADHD in first-time diagnostic evaluations, it is crucial to extend these methods to this population. This includes exploring the underlying network and factor structure of PVTs, alongside SVTs, and other clinical measures.

The present study aimed to clarify the conceptual similarities and distinctions between PVTs, CPTs, and ADHD symptom report scales, including embedded SVTs, addressing a gap in the understanding of the validity framework in adult ADHD assessments. To test this, we employed a combination of network and factor analysis techniques to examine both the interrelationships and factorial structures of three PVT measures among a large set of neuropsychological measures. This analysis was based on data from a large clinical referral sample of adults with suspected ADHD. The comprehensive test battery included ADHD symptom self- and observer reports, two embedded SVTs, a CPT, and three stand-alone PVTs. Similarly to [102] ([102]), we examine the presence of a three-factor structure of validity assessments. This structure encompasses (1) performance validity measures, (2) continuous performance measures, and (3) symptom reports. SVTs are expected to load onto the symptom report factor, due to their close relationship with reported ADHD self-reported symptoms. Additionally, we examined potential influential factors of PVT results. For this, we considered a range of demographic (age, sex, education level) and clinical characteristics (ADHD diagnosis, ADHD symptom reports, comorbidity, cognitive deficit). As freestanding PVTs are designed to be easily achievable, we expect that these demographic and clinical factors should have little to no influence on PVT pass/failure rates.

## 2. Materials and Methods

### 2.1. Procedure and Participants

All participants were recruited between 2022 and 2024 from the adult ADHD outpatient clinic at the Department of Psychology, Philipps University Marburg, Germany. See [58] ([58]; manuscript submitted for publication) for details and previous research on subsets of this dataset. All clients provided written informed consent for their data to be used for research purposes. The local Ethical Review Committee approved the use of routine data from the outpatient clinic.

Individuals were referred for a thorough diagnostic assessment of ADHD due to concerns raised by medical professionals, significant others, or themselves. The diagnostic assessment was based on the empirically informed guidelines for first-time ADHD diagnoses in adults ([90]). Clients were assessed by board-certified clinical psychologists based on provided and reported clinical history. The standard diagnostic examination included (1) semi-structured interviews, the diagnostic interview for ADHD in adults (DIVA-5; [59]), and the structured clinical interview for DSM-IV (SCID-I; [106]) for assessing potential comorbidity or other primary disorder explaining symptoms; and (2) the Conners Adult ADHD Scale (CAARS-L; [15]; [18]) employed both as self (CAARS-L:S) and observer (CAARS-L:O) questionnaires for assessing ADHD psychopathology. Alongside these clinical tools, clinicians also assessed the impact of ADHD symptoms on various areas of daily functioning, such as academic challenges, workplace difficulties, participation in risky behaviors, and substance abuse issues.

Neuropsychological assessment of ADHD core symptoms was done with the Quantified Behaviour Test (Qb+; [101]), a continuous performance test (CPT) with actigraphy of head movement that served as objective and normed measures of attention, impulsivity, and motor activity. This was followed by three PVTs, namely the Amsterdam Short-Term Memory Test (ASTM; [88]), the Medical Symptom Validity Test (MSVT; [40]), and the Groningen Effort Test (GET; [35]; see below for details). For this study, two embedded SVTs, the Inconsistency Index and Infrequency Index (CII), were calculated separately ([15]; [18]; [94]). These were based on the CAARS-L:S questionnaire, providing additional insight into infrequent and inconsistent response patterns.

Validity test results were considered in the diagnostic process, particularly in cases with ambiguous findings. If an ADHD diagnosis was strongly supported by other assessment measures and the validity concerns were explainable, the diagnosis remained unaffected. However, in cases with inconclusive or inconsistent results, validity findings played a decisive role in diagnostic determination.

Data was collected from 479 clients, which included performance data from PVT, SVT, CPT, and diagnostic assessments. Clients were excluded from the analyses due to incomplete cases (*N* = 17) and administrative errors (*N* = 1). The final sample included 461 participants. Descriptive statistics are depicted in Table 1. Overall, individuals’ average age was 33.2 years (SD = 10.4), and more than half of them were women (58.1%, *N* = 268). The participants’ educational backgrounds varied, including individuals not reported/without a school-leaving certificate (0.9%, *N* = 4) and general education (basic/secondary school degree; 32.5%, *N* = 150), with the majority having advanced education (grammar school/university degree; 66.6%, *N* = 307). Among all adults who underwent ADHD diagnostic assessment, 84.2% (*N* = 388) received a diagnosis post evaluation. Within this sample, 26.0% (*N* = 101) of adults reported at least one (-up to three) comorbid mental disorder(s).

### 2.2. Measures

#### 2.2.1. Amsterdam Short-Term Memory (ASTM) Test

The ASTM test functions as a performance validity test (PVT), detecting negative response bias and assessing insufficient effort or motivation during psychological evaluations ([88]). It is a memory test that uses a forced-choice recognition method, designed to assess short-term memory and attention. The ASTM test involves displaying five semantically related words for eight seconds, which participants are instructed to read aloud and memorize. Afterwards, they complete a simple arithmetic problem. Subsequently, five words are shown again, including three from the original set. Participants must identify the three words they saw previously. The used cut-off values and interpretation guidelines follow the recommendations outlined in the test manual ([88]). [88] ([88]) reported high internal consistency (Cronbach’s α = 0.90) for various samples. For experimental simulants with neurological impairments, sensitivity for lack of motivation was 91% and specificity was 89%. Patients with neurological disorders, i.e., brain tumors and concussion, rarely have difficulties handling this test, given that they do not have serious cognitive deficits.

#### 2.2.2. Medical Symptom Validity Test (MSVT)

The MSVT is a brief computerized measure of cognitive effort and memory ([40]). According to the author, the MSVT was designed to shorten the administration of the test’s parent version, the Word Memory Test (WMT; [39]). Like the WMT, the MSVT displays common word pairs over two consecutive trials; Immediate Recall (IR), Delayed Recall (DR), and Consistency Scores (CNS) were used to evaluate performance validity based on the test’s manual criteria. The Genuine Memory Impairment profile was not used, as it is cautioned against for individuals who do not exhibit significant functional decline ([78]). Given the nature of our sample, such a decline was not expected, and its assessment was beyond the scope of the present study. The classification of values for IR, DR, or CNS as PVT failure follows the guidelines outlined in the test manual ([40]). The MSVT has shown adequate sensitivity and specificity in detecting invalid performance. For example, sensitivity predicting the WMT was 0.50 to 0.62 ([41]).

#### 2.2.3. Groningen Effort Test (GET)

The GET is an attention-based validity assessment, focusing on detecting underperformance ([35]). It is an Embedded Figure Test (EFT), meaning the GET consists of a simple geometric figure embedded in a more complex one ([36]). During the test, participants need to indicate on a computer screen whether the simple target figure is embedded within a complex figure. Both figures are displayed simultaneously, requiring participants to indicate the presence or absence of the embedded figure by pressing a button. The interpretation of index values and error rates for assessing cognitive performance in individuals with ADHD follows the guidelines provided in the test manual ([35]). The GET index score is calculated by combining processing speed and accuracy measures within each block of trials. In addition, the total number of errors is calculated. Both the index score and total number of errors were used in the current study to indicate noncredible performance. Analogue studies indicate that the GET successfully differentiates simulated malingered ADHD from genuine ADHD with high sensitivity (68%) and specificity (91%) when using a suggested cut-off score for total errors ([35]).

#### 2.2.4. Quantified Behaviour Test Plus (Qb+)

The QbTest+ ([77]) is a computer-based continuous performance test (CPT), measuring sustained attention with a one-back working memory task. Essentially, participants are required to recall the same object (red or blue circles and squares) in shape and color. A response key is to be pressed when the current stimulus matches the previous one in both attributes ([24]; [101]). Participants are instructed to store and compare stimuli in working memory ([64]), with a target-to-nontarget stimulus ratio of 25:75. While performing the CPT, participants sit 1 m from a monitor with an infrared head movement tracking camera and use a handheld responder. The camera captures movement during task execution, at a frequency of 50 samples a second with a spatial resolution of 1/27 mm per infrared camera unit. Normative data have been gathered from 1307 individuals between 6–60 years of age for both versions of the test (QbTest 6–12 and Qb+) with an even age and gender distribution ([101]). Q-scores are derived for hyperactivity, inattention, and impulsivity. They are interpreted similarly to Z-scores with a mean of 0 and a standard deviation of 1. A Q-score of >1.5 is observed as an atypical result ([46]; [101]). The Qb+ reports nine parameters that are categorized into activity and CPT measures. Of these, we report four specific CPT measures in total: (1) Reaction Time Q-score (RTQ) in milliseconds, reflecting the average time for correct responses, offering insights into both information processing latency and motor response speed; (2) RT Variation (RTVarQ) is calculated using the standard deviation of the mean of correct response times and is a measure of the participant’s inconsistency in response times; (3) Omission Error score (OmissQ) reflects the total number of missed targets; (4) Commission Error score (ComissQ) is the number of false hits. In the present study, Q-scores that are normed for age and sex were utilized for the listed parameters. Sensitivity and specificity are within the acceptable range ([9]).

#### 2.2.5. Conners Adult ADHD Rating Scales (CAARS-L:S; CAARS-L:O)

The CAARS-L:S and CAARS-L:O (German version) assess ADHD symptoms in adults aged 18 and older. The long version consists of 66 items rated on a Likert-type scale ranging from 0 (not all all/never) to 3 (very much/very frequently). Due to statistical restrictions, only 42 items were included in the original factor analysis, revealing four factors: inattention/memory problems, hyperactivity/restlessness, impulsivity/emotional lability, and problems with self-concept ([18]). These were further confirmed in the German version ([16]; [17]). Both rating scales have been considered as reliable and cross-culturally valid tools for assessing current ADHD symptoms in adults ([17]). Norm scores are calculated based on sex and age. For the present study, the four core symptoms of inattention, hyperactivity, impulsivity, and self-concept problems were included.

In addition, two embedded SVTs were obtained from the self-rated questionnaires. Firstly, the CAARS-L:S—Inconsistency Index was computed, indicating the probable occurrence of careless responding for any score > 8. The manual suggests inspecting this score first to detect problems with internally consistent responding, which may affect the validity of the CAARS. Secondly, the CII ([94]) was calculated by summing twelve item scores, which were shown to be infrequently reported by healthy controls and adults with credible ADHD. Any score of the CII > 21 indicates an elevated, noncredible symptom report. Early studies of the CII showed sensitivity estimates ranging from 17% to 19% ([8]; [19]; [20]; [36]) to 86% ([80]) and 95% ([104]).

### 2.3. Statistical Analyses

All analyses were performed using RStudio (Version 4.3.3; [83]). Less than 6% of clients (25 out of 461) had missing CAARS-L:S items, accounting for 5.4% of the data. In line with the CAARS-L:S manual, ≤5 missing items is considered acceptable ([15]). Missing items were handled using multiple imputation with the mice package ([84], [85]).

To examine the interrelationships among the given variables (PVTs, Qb+, CAARS-L:S/O, including its SVTs), a heatmap analysis of the Pearson correlation matrix was conducted. This step ensured sufficient common variance for factor extraction and helped identify potential multicollinearity. Furthermore, a network analysis was then performed to explore structural connections among the variables. To test the hypothesized factor structure of PVTs in adults with suspected ADHD, confirmatory factor analysis (CFA) was conducted. Due to insufficient support from the CFA for the hypothesized structure, post hoc exploratory factor analyses (EFAs) were subsequently performed to further explore the latent dimensions. Lastly, binary logistic regression models were conducted to examine factors influencing PVT pass and fail outcomes.

#### 2.3.1. Descriptive Statistics and Preliminary Heatmap Correlations

Test performance (PVTs, Qb+) and clinical information (CAARS-L:S/O, including SVTs) are presented descriptively for all individuals. PVT results are shown as positive results, as the interpretation of PVT outcomes is commonly conducted dichotomously (positive indicating failure, negative indicating pass). Furthermore, ADHD symptom reports and Qb+ measures are presented using descriptive statistics and interpreted based on test norms, i.e., showing the proportion of individuals with “below average” self-reports. “Below average” is defined by [42] ([42]) as a percentile rank (PR) ≤ 8 on tests and a PR ≥ 92 on self-report scales.

To explore the interrelationships and underlying factors among the given variables, we first applied a heatmap analysis to the Pearson correlation matrix. Visualizing the correlations among PVTs, Qb+, and CARRS-L:S/O (including SVTs) in a heatmap was essential for evaluating the suitability of these variables in identifying a potential PVT factor. This preliminary step allowed for the assessment of the strength and direction of intercorrelations, ensuring that the variables shared sufficient common variance required for factor extraction. Additionally, the heatmap helped detect multicollinearity, which could hinder meaningful PVT factor identification. It also confirmed that the relationships between these diverse measures were robust enough for both network and factor analysis.

#### 2.3.2. Network Analysis

A network analysis was conducted to visually and structurally explore the interrelationships of all variables, including PVTs (MSVT, ASTM, GET), Qb+, and CAARS-L:S/O (including SVTs). The network analysis consists of three core analytical steps: (1) estimating the network; (2) assessing node centrality; (3) evaluating accuracy and stability. First, network estimation visually represents the relationships between variables, which can be quantified using partial correlation to measure the strength of these connections. The networks were constructed using the R package “bootnet” (Version 1.6; [25]; [26]), employing Graphical Gaussian Models with the Least Absolute Shrinkage and Selection Operator (EBIC; [32]). To enhance interpretability and prevent spurious connections, the “Least Absolute Shrinkage and Selection Operator” (LASSO; [98]) regularization algorithm was employed. Network visualization relied on the Fruchterman–Reingold algorithm ([33]). In the graph produced by the Fruchterman–Reingold algorithm, nodes with stronger connections are positioned closer to each other, and edges between nodes with higher absolute coefficients are depicted with thicker and more saturated colored lines. A detailed description of the analytic procedures is available in [21] ([21]) and [25] ([25]). As our data was not normally distributed, Spearman correlations were utilized as input ([52]).

Next, we evaluated the centrality of each item, which indicates how connected an item is to all other items in the network. We used expected influence, a centrality measure that sums the connections for each node. To compute and visualize the expected influence, we employed the centrality, centralityTable, and centralityPlot functions from the “qgraph” package ([26]). These analyses identified whether items in the network clustered together in a particular manner.

Lastly, we evaluated the precision of edge weights and the consistency of the node centrality order. The accuracy of edge weights was evaluated using bootstrapping to compute the 95% confidence interval (CI) for each edge. Smaller CIs suggest greater accuracy in the ordering of edges within the network. Node centrality stability was assessed using the correlation stability (CS) coefficient. Based on the simulation approach outlined by [25] ([25]), CS coefficients above 0.25 indicate moderate stability, while those above 0.5 indicate strong stability. These were performed using the R package “bootnet” ([25]; [26]).

#### 2.3.3. Confirmatory Factor Analysis

Furthermore, a confirmatory factor analysis (CFA) was employed to assess the expected factor structure of PVTs in adults with suspected ADHD. CFA validates the pre-established hypothesized relationships between observed indicators and latent factors, providing evidence of constructs in theory-based instrument construction and development ([57]). As our observed variables (20 items) include both count and continuous data (RTQ, RTVarQ, OmissQ, ComissQ), and considering our sample size (<500), we used the Robust Maximum Likelihood Estimator (MLR). The MLR was chosen as it adjusts for nonnormality and is well-suited for managing mixed-type data ([55]; [65]).

To evaluate the overall goodness-of-fit of the model we employed standard indices, including standard chi-square statistics (*χ*^2^/*df*), the robust root mean square error of approximation (RMSEA), the Robust Confirmatory Fit Index (CFI), and the Robust Tucker–Lewis Index (TLI) ([12]; [51]). Model–data fit is considered acceptable when χ^2^/*df* ranges from 2 to 5, the RMSEA is below 0.06, and the CFI and TLI exceed 0.80. A good fit is achieved when χ^2^/*df* is close to 1, and both the CFI and TLI exceed 0.95 ([50]). In line with previous research, suggesting the PVT as a latent factor, we specified three models with one (all items), two (Test Performance, Symptom Report), and three (Performance Validity, CPT, Symptom Report) latent factors.

#### 2.3.4. Multiple Logistic Regression Models

To examine factors influencing the likelihood of passing or failing each PVT, a series of binary logistic regression models was conducted. Logistic regression with a binary outcome (pass/fail) was preferred over linear regression, as the goal was to analyze clinically relevant outcomes for PVTs. The analytical approach depended on whether a robust PVT factor structure emerged. If a robust factor structure was identified, a composite PVT factor would be computed by weighting the individual PVTs according to their respective factor scores. Subsequent analyses would then focus on the composite PVT factor. If no robust factor structure emerged, each PVT (ASTM, MSVT, GET) would be analyzed separately. The outcome for each PVT would be binary (pass/failure), based on the cut-off criteria outlined in the respective test manuals ([35]; [40]; [88]).

Predictor variables included demographic factors (age, sex, and education level) and clinical data (ADHD diagnosis, ADHD core symptoms, comorbidity, and cognitive deficit). These predictors were selected because they represent key characteristics relevant to adult ADHD and provide insights into individual differences in PVT performance. ADHD core symptoms (inattention, hyperactivity, impulsivity, and self-concept problems) were classified based on the CAARS-L:S/O questionnaires, with clients scoring > 65 (T-scores) considered to exhibit clinically significant symptoms. The cognitive deficit variable was derived from the Qb+ assessments, with a Q-score > 1.5 on any of the four CPT measures (RTQ, RTVarQ, OmissionQ, CommissionQ) considered indicative of a deficit.

In the absence of a factor structure, odds ratios (ORs) for each predictor variable were computed by exponentiating the model coefficients for easier interpretation. Logistic regression analyses were performed separately for each PVT (ASTM, MSVT, and GET) using the “*glm*” package in R. Model fit was assessed using Nagelkerke’s R^2^ and the likelihood ratio test to evaluate the explanatory power of the logistic regression models ([22]; [72]).

## 3. Results

### 3.1. Descriptive Statistics and Preliminary Correlations

Table 2 presents the descriptive statistics for the PVTs, Qb+, CARRS-L:S/O (including SVTs), along with the proportion of individuals who obtained positive results on each PVT subtest separately. The proportion of conspicuous performance differed by subtest, with 1.7% (MSVT), 10.2% (GET), and 15.2% (ASTM). Inconsistent responding in the CAARS was observed in 23.2% of the individuals, while 23.6% gave infrequent responses (CII). The analysis of the CAARS-L:S/O questionnaires shows that a majority of participants scored PR > 92% on at least one ADHD-related scale, particularly self-reports. Among all CAARSs, the highest number of individuals had elevated scores on “Inattention”, followed by “Self-Concept Problems”, “Impulsivity”, and “Hyperactivity”. Additionally, more individuals exhibited high scores in the self-report scales compared to the observer ratings.

Before analyzing the relationships and factors among the parameters PVTs (MSVT, ASTM, GET), SVTs (CAARS-L:S—Inconsistency, CII), and CAARS-L (CAARS-L:S, CAARS-L:O), all variables were z-transformed using a robust Z-score based on the current data sample. This transformation ensured that each variable had a mean of zero and a standard deviation of one. It was performed to mitigate potential effects of scale discrepancies between variables and to prevent estimation issues caused by differences in response variances. Q-based norm scores (adjusted for age and gender) from the Qb+ test were entered into the analysis.

Figure 1 displays the heatmap of the correlation matrix for each PVT, Qb+ test, and CARRS-L:S/O (including SVTs). ASTM showed weak correlations with MSVT subscales (IR, DR, CNS; *r* = 0.14 to 0.22). GET variables (Errors, Index) did not correlate significantly with ASTM (*r* = −0.18 to −0.31) or MSVT (*r* = −0.08 to −0.25), indicating weak to negligible associations. Correlations between PVT measures and CAARS symptom scales were weak to negligible (*r* < 0.30). Notably, both the CAARS Infrequency Index (CAARS_CII) and the CAARS Inconsistency scale (CAARS_Incon) showed weak correlations with CAARSs and PVT measures. However, CAARS_CII was moderately correlated with CAARS self-reported scales “Inattention” and “Impulsivity” (*r* = 0.56 to 0.78). Lastly, Qb+ test measures (RTQ, RTVarQ, CommQ, OmissQ) displayed weak correlations with CAARS symptom scales, SVT, and PVT measures (*r* < 0.30).

### 3.2. Network Analysis

The item–network estimation of the 20 PVT, SVT, and Qb+ measures, along with ADHD symptom reports, is presented in Figure 2. The network consists of 75 edges (45 positive, 30 negative), showing multiple connections between the different measures (nodes). The network structure reveals connections between validity measures (PVT: nodes 1–6; SVT: nodes 11–12) with other ADHD assessment domains (QB+: nodes 7–10; CAARS: nodes 13–20), rather than forming a separate PVT subgroup. This indeed goes hand-in-hand with the results of the correlation matrix. SVT nodes (11–12) display strong positive connections with ADHD symptom scales (12–20). Node 11 exhibits stronger connections across impulsivity scales (15, 19), whereas node 12 is more linked to self-concept (16). In contrast, the PVT nodes exhibit distinct patterns of connectivity. Nodes 1–4 show minimal associations with ADHD symptom scales but display some connections with each other. Nodes 5–6 show slightly stronger relationships with Qb+ measures (7–10).

To further explore the centrality of individual nodes within the network, an expected influence analysis was conducted (Figure 3). ADHD reported “self-concept” scales exhibited the highest expected influence. Whilst SVTs displayed moderate expected influence, PVTs demonstrated the lowest expected influence. The edge weight accuracy estimation revealed a moderate confidence interval, indicating that the rank order of edge weights was accurately estimated (Figure 4). Furthermore, the node centrality estimation revealed CS coefficients of 0.751 for the expected influence and strength of the entire sample, indicating that the order of these centrality measures was highly stable. Given the observed network structure and stability, a CFA further assessed the constructs among validity and ADHD measures.

### 3.3. Confirmatory Factor Analysis

See Table 3 for the item allocation to the respective factor structures. Table 4 presents the fit indices for the one-, two-, and three-factor CFA models. The approximate model fit indices (χ^2^/df, CFI, RMSEA, and TLI) displayed poor fit for one-, two-, and three-factor models. This suggests that while a structure was present, it differed from the hypothesized factor structure of (1) performance validity measures, (2) continuous performance measures, and (3) symptom reports, including SVTs. For the three-factor model, the chi-square statistic was χ^2^(1550.4) with a χ^2^/df ratio of 9.28, exceeding the acceptable range of 2 to 5 ([50]). The RMSEA was 0.136, which is above the commonly accepted threshold of 0.06 for good fit. Additionally, the CFI (0.669) and TLI (0.623) were below the recommended 0.80 threshold for acceptable model fit. Similar patterns of poor fit were observed for the two-factor model (χ^2^/df = 11.5, RMSEA = 0.154, CFI = 0.570, TLI = 0.517) and the one-factor model (χ^2^/df = 16.7, RMSEA = 0.19, CFI = 0.34, TLI = 0.27).

### 3.4. Post Hoc Analysis: Exploring Alternative Model Fits

Due to the poor fit of the pre-specified one-, two-, and three-factor models in the CFA, an exploratory factor analysis (EFA) was subsequently conducted. This analysis aimed to identify potential alternative factor structures underlying the PVTs, CPTs, and symptom reports (including SVTs). Bartlett’s Test of Sphericity was significant χ^2^(190) = 4427.2, *p* < 0.001, and the overall Kaiser–Meyer–Olkin (KMO) measure was acceptable at 0.61, supporting factorability ([6]; [53], [54]). Factors were extracted using maximum likelihood estimation with oblimin rotation, and the number of factors was determined via parallel analysis ([29]). Four factors were retained based on the scree plot and parallel analysis (Appendix A). Factor 1 included the majority of CAARS subscales (self- and observer ratings) along with the embedded SVT (CII). Factor 2 was defined primarily by Qb+ variables. Factor 3 was characterized by strong loadings from the MSVT subscale. Factor 4 was defined by GET variables.

### 3.5. Multiple Logistic Regression Models

Since the CFA analysis did not reveal a common performance validity factor, each PVT was analyzed separately in terms of pass/fail likelihood. Multiple logistic regression models were computed to examine the predictors of PVT outcomes. Results are presented in Table 5. Odds ratios (ORs) and statistically significant levels are reported for each predictor.

The logistic regression model predicting the likelihood of passing the ASTM test (PVT 1) was statistically significant, χ^2^(12) = 26.8, *p* = 0.008. The model explained 10.7% of the variance in the likelihood of passing the ASTM test (R^2^ = 0.107). Cognitive deficits were significantly associated with lower odds of passing the ASTM test (*β* = −1.23, *SE* = 0.374, OR = 0.29, *p* < 0.001). Additionally, education was associated with greater odds of passing (*β* = 0.302, *SE* = 0.132, OR = 1.35, *p* < 0.05). Among CAARS-L:O results, the impulsivity ADHD scale was a significant predictor (*β* = 0.987, *SE* = 0.360, OR = 2.68, *p* < 0.01), indicating that higher impulsivity scores were associated with increased odds of passing the ASTM test. Conversely, the impulsivity reported symptom from the CAARS-L:S was negatively associated with passing the test (*β* = −0.791, *SE* = 0.402, OR = 0.45, *p* < 0.05). A further logistic regression was conducted to examine the factors influencing the likelihood of passing the MSVT (PVT 2). The model was not statistically significant, χ^2^(12) = 18.5, *p* = 0.101. Finally, a logistic model was used to predict the likelihood of passing the GET (PVT3). The model was not statistically significant, χ^2^ (12) = 20.5, *p* = 0.059.

## 4. Discussion

The primary objective was to explore the shared and unique structural features between PVTs, CPTs, and ADHD symptom scales (including SVTs), using data from a large outpatient sample of adults with suspected ADHD. To address this aim, we initially hypothesized that the three PVTs (ASTM, MSVT, GET) would load onto a common latent construct. However, the results do not support this hypothesis.

Descriptive statistics revealed differences in positive PVT results, with 1.7% (MSVT), 10.2% (GET), and 15.2% (ASTM). This variability broadly resembles findings from prior research ([47]; [67]) and underscores recommendations to administer multiple PVTs ([10]). In contrast, SVT elevated scores were more consistent, with 23.2% of individuals showing inconsistent and 23.6% infrequent responses (CII; [36]).

Initial correlational patterns suggested weak relationships between PVTs, ADHD symptom scales (including SVTs), and CPT measures, hinting at structural independence between test modalities. A visual inspection of the network analysis further supported this, revealing four emergent item communities. The ASTM test remained largely independent, with only weak connections to the GET and CPT variables. The MSVT variables formed a cohesive cluster. A similar pattern of internal cohesion and external separation was observed for the GET and CPT variables. As expected, the subscales of the CAARS exhibited strong interrelationships in both self- and observer ratings, closely clustered with SVTs ([18]; [94]). This pattern is further supported by the finding that ADHD self-concept scales displayed the highest, SVTs moderate, and PVTs the lowest centrality.

Further insights into the PVT structures were gained through factor analytic approaches. As a first approach, we conducted a CFA to test a theoretically driven three-factor model distinguishing (1) performance validity tests, (2) continuous performance measures, and (3) ADHD symptom reports (including SVTs). Two further competing models were tested, namely a one-factor solution and a two-factor model distinguishing test performance (ASTM, MSVT, GET, QB+) from symptom reports (CAARS-L self- and observer ratings, including SVTs). The latter model was theoretically motivated by the assumption that both PVT and CPT measures reflected aspects of test performance, whereas CAARS ratings (including SVTs) capture symptom reports. Contrary to our hypothesized structure and competing model specifications, all tested CFA models (one-, two-, and three-factor) showed poor fit across all indices. Following the poor CFA model fits that falsified the assumption of a common PVT factor, we further explored the latent structure of the measures through an EFA. Unlike the CFA, the EFA supported a four-factor solution, offering clearer differentiation among the test modalities. Notably, the PVTs did not form a single coherent factor but rather loaded onto separate factors, reflecting their distinct test characteristics. This structure complements the item communalities in the network analysis and provides further evidence against a unified PVT construct.

The factor analyses reveal an important distinction, while contradicting previous research by [7] ([7]) and [102] ([102]). These two studies reported a distinct performance validity factor, primarily based on memory-focused PVTs, which tend to share similar demands ([69]; [70]). Our battery included a broader range, such as the attention-based GET and performance measures like Qb+, introducing more variability. This diversity likely weakened correlations among tests and reduced the coherence expected in the CFA. In this context, the domain specificity hypothesis proposed by [27] ([27]) offers a useful theoretical lens. It suggests that PVT outcomes reflect specific cognitive domains and task characteristics, rather than representing a single, unified construct. For example, PVTs assessing sustained attention (e.g., GET) and those assessing verbal memory (e.g., ASTM, MSVT) engage different cognitive systems, which may result in distinct response bias patterns within the same individual. Importantly, PVTs are not intended to measure underlying cognitive ability ([49]; [103]). Instead, they help contextualize actual test performance and should remain largely independent of the cognitive functions they reference. This perspective explains why correlations among different PVTs are often moderate ([82]) and supports the widely accepted recommendation to administer multiple PVTs targeting distinct cognitive domains ([10]). Our findings add empirical support to this view and contradict [61]’s ([61], [62]) claim of PVT collinearity by showing that each test evaluates distinct cognitive processes. From this viewpoint, the more unified validity structures reported by [7] ([7]) and [102] ([102]) are less convincing. Their findings challenge the rationale for using multiple PVTs and underestimate the importance of a domain-sensitive approach to performance validity assessment in diverse clinical populations.

The second objective of this study addressed the influence of demographic and clinical factors on PVT performance, further emphasizing the independent functioning of each PVT. Whilst the MSVT and GET followed our expectation and showed no influence of demographic or clinical factors, the ASTM test did, as cognitive deficits, education level, and impulsivity were significant predictors of ASTM performance. Cognitive deficits were linked to lower odds of passing the ASTM test, consistent with prior research ([23]; [69]). Education level was a positive predictor, aligning with [93] ([93]), who also reported better performance in individuals with higher educational attainment. Impulsivity showed an unexpected, mixed relationship. Higher impulsivity on the CAARS-L:O was associated with passing, whereas impulsivity on the CAARS-L:S had the opposite effect. These findings align with [27]’s ([27], [28]) view that performance validity exists along a continuum rather than a strict pass/fail outcome. According to [27] ([27]), engagement, performance, and cognitive functioning influence where individuals fall on this spectrum. Cognitive deficits, education, and impulsivity may affect how people engage with the ASTM test, leading to varied performance. However, this sensitivity to demographic and clinical factors may increase the risk of false positives. Some individuals could be misclassified as underperforming despite genuine cognitive difficulties or other traits. This limitation may reduce the ASTM test’s usefulness across diverse clinical groups. By contrast, the MSVT and GET appear more robust and less influenced by demographic and clinical factors, thus assessing the actual performance demonstrated during testing rather than being confounded by individuals’ underlying cognitive potential.

More generally, many PVTs have demonstrated low to modest sensitivity at standard cut-offs in ADHD contexts ([79]; [95]). This supports [79]’s ([79]) recommendation to adapt PVT procedures and interpretation specifically for individuals with ADHD. Therefore, PVTs should be interpreted in conjunction with broader clinical data, rather than in isolation. This aligns with the current practice guidelines, which advocate for a comprehensive, multi-method approach to evaluating performance validity ([89]; [97]).

### 4.1. Limitations

Several limitations should be considered when interpreting the findings of this study. The current sample is relatively homogenous in terms of gender (58.1% female) and education (66.6% with at least grammar school). This limits the generalizability of the results. Future analyses involving more diverse samples and subgroup analyses could help clarify which demographic variables influence ADHD diagnoses and validity.

On this note, while the use of a naturalistic clinical referral sample strengthens the study’s ecological validity, it also brings challenges in confidently identifying true ADHD cases. Variability in symptom presentation and the potential for distorted reporting or secondary gain can complicate diagnostic clarity. So, incorporating external validation criteria in future research may help address these concerns ([38]).

Furthermore, the SVTs included in this study were embedded within ADHD symptom measures, specifically the CAARS self-report scale. While this allows for context-sensitive assessment of symptom validity, the lack of stand-alone SVTs limits the ability to disentangle symptom exaggeration from genuine symptom expression. Future analyses should therefore consider integrating stand-alone SVTs.

### 4.2. Implications for Future Research

Our findings reinforce the need to move beyond a unidimensional view of performance and potentially symptom validity. Accordingly, future studies should test multidimensional models that account for domain-specific underperformance (e.g., attention- vs. memory-related validity). Recent work by [31] ([31]) supports this approach, showing that combining embedded indicators across cognitive domains improves detection of underperformance in ADHD evaluations. These findings underscore the importance of task characteristics and cognitive domain in shaping performance validity.

As a complementary approach, future research may benefit from adopting person-centered analytic strategies to further characterize heterogeneity in performance validity outcomes. This approach could identify subgroups with distinct cognitive profiles, similar to those found by [13] ([13]). In doing so, it may uncover distinct patterns across validity indicators, shedding light on ADHD’s neurocognitive heterogeneity.

Finally, future research should consider [27]’s ([27], [28]) conceptualization of performance validity as a continuum. This may be particularly relevant in ADHD, where engagement often varies with task demands. Modelling performance and symptom validity on a continuum could clarify how levels of underperformance relate to cognitive deficits, impulsivity, or education, thus improving the sensitivity of validity assessment in ADHD by capturing individual and task-related variability more accurately.

## 5. Conclusions

This study highlights the multifaceted nature of PVTs, CPTs, and ADHD symptom reports (including SVTs) in an outpatient adult ADHD population. Our findings suggest that PVTs (ASTM, MSVT, GET) reflect domain-specific processes, with performance influenced by the cognitive domain (e.g., sustained attention versus memory), rather than a unified latent construct. This supports a multidimensional view of performance validity and highlights the role of task-specific and individual factors. Notably, ASTM test performance was sensitive to cognitive deficits, education level, and impulsivity, underscoring the importance of interpreting PVT outcomes within the broader clinical context and along a behavioral continuum. These findings further support the need for ADHD-specific procedures and adjusted cut-offs to improve the accuracy of performance validity assessment in this population. These insights also carry important implications for forensic contexts, emphasizing that performance validity measures offer an additional behavioral indicator essential for reducing misclassifications in legal evaluations.

Future research should model performance validity as a multidimensional, continuous construct. Person-centered approaches may further clarify how cognitive and behavioral profiles shape test performance in ADHD, enhancing the precision and utility of validity assessments.

## Figures and Tables

**Figure 1 behavsci-15-01108-f001:**
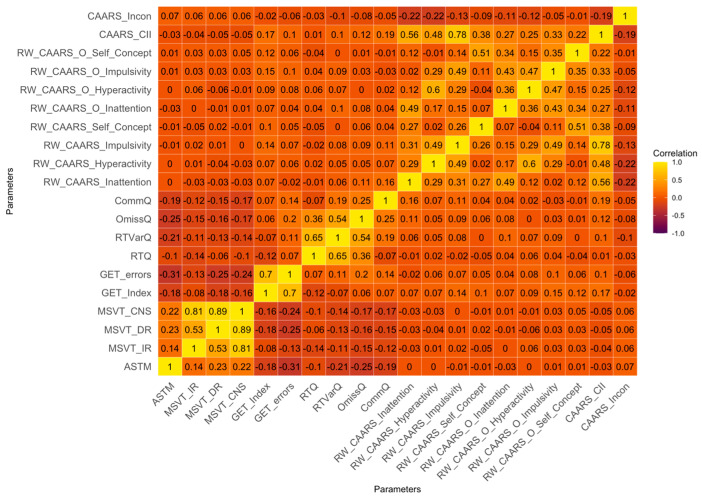
Heatmap of the correlation matrix for each PVT, SVT, Qb+ test, and self-reported ADHD symptoms (*N* = 461). Note. ASTM = Amsterdam Short-Term Memory test; MSVT_IR = Medical Symptom Validity Test (Immediate Recall); MSVT_DR = Medical Symptom Validity Test (Delayed Recognition); MSVT_CNS = Medical Symptom Validity Test (Consistency Scores); GET_Index = Groningen Effort Test (Index); GET_errors = Groningen Effort Test (Errors); RTQ = Reaction Time Q-score (Qb+); RTVarQ = Reaction Time Variation Q-score (Qb+); OmissQ = Omission Error Q-score (Qb+); CommQ = Commission Error Q-score (Qb+); RW_CAARS_Inattention = CAARS-L:S (Inattention); RW_CAARS_Hyperactivity = CAARS-L:S (Hyperactivity); RW_CAARS_Impulsivity = CAARS-L:S (Impulsivity); RW_CAARS_Self_Concept = CAARS-L:S (Self-Concept); RW_CAARS_O_Inattention = CAARS-L:O (Inattention); RW_CAARS_O_Hyperactivity = CAARS-L:O (Hyperactivity); RW_CAARS_Impulsivity = CAARS-L:O (Impulsivity); RW_CAARS_O_Self_Concept = CAARS-L:O (Self-Concept); CAARS_CII = CAARS-L:S (Infrequency Index); CAARS_Incon = CAARS-L:S (Inconsistency Index). Heatmap follows a gradient, where yellow represents a strong positive correlation (r ≈ +1), dark brown/purple represents a strong negative correlation (r ≈ −1), and darker orange shades indicate weaker/no correlation (r ≈ 0).

**Figure 2 behavsci-15-01108-f002:**
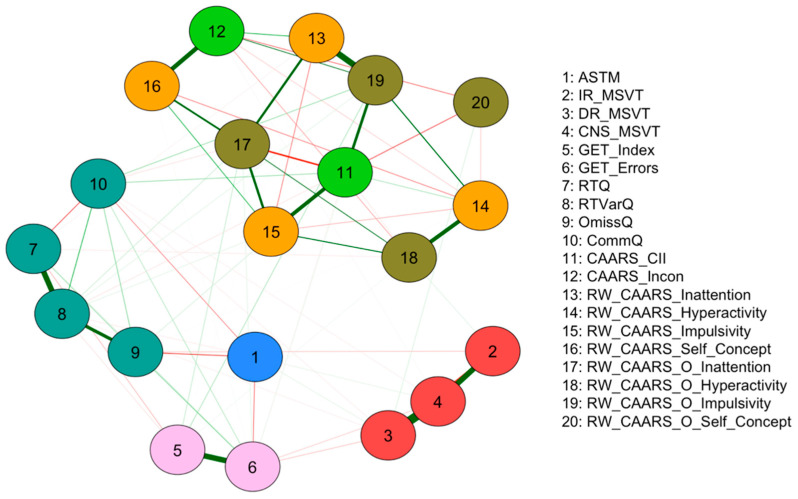
Network analysis of PVT, SVT, and Qb+ measures in relation to ADHD symptoms in the full sample (*N* = 461). Note: nodes represent the PVT (ASTM, MSVT, GET), QB+ (RTQ, RTVarQ, OmissQ, CommQ), and SVT (CAARS_CII, CAARS_Incon) variables and the CAARS_S/O (Inattention, Hyperactivity, Impulsivity, Self-Concept) scales. Variables are color-scaled in the aforementioned groups. The edges connecting the nodes depict regularized partial Spearman correlations. Thicker and more vividly colored edges indicate stronger absolute correlations. Green edges display positive correlations; red edges display negative correlations. ASTM = Amsterdam Short-Term Memory test; MSVT_IR = Medical Symptom Validity Test (Immediate Recall); MSVT_DR = Medical Symptom Validity Test (Delayed Recognition); MSVT_CNS = Medical Symptom Validity Test (Consistency Scores); GET_Index = Groningen Effort Test (Index); GET_errors = Groningen Effort Test (Errors); RTQ = Reaction Time Q-score (Qb+); RTVarQ = Reaction Time Variation Q-score (Qb+); OmissQ = Omission Error Q-score (Qb+); CommQ = Commission Error Q-score (Qb+); RW_CAARS_Inattention = CAARS-L:S (Inattention); RW_CAARS_Hyperactivity = CAARS-L:S (Hyperactivity); RW_CAARS_Impulsivity = CAARS-L:S (Impulsivity); RW_CAARS_Self_Concept = CAARS-L:S (Self-Concept); RW_CAARS_O_Inattention = CAARS-L:O (Inattention); RW_CAARS_O_Hyperactivity = CAARS-L:O (Hyperactivity); RW_CAARS_Impulsivity = CAARS-L:O (Impulsivity); RW_CAARS_O_Self_Concept = CAARS-L:O (Self-Concept); CAARS_CII = CAARS-L:S (Infrequency Index); CAARS_Incon = CAARS-L:S (Inconsistency Index).

**Figure 3 behavsci-15-01108-f003:**
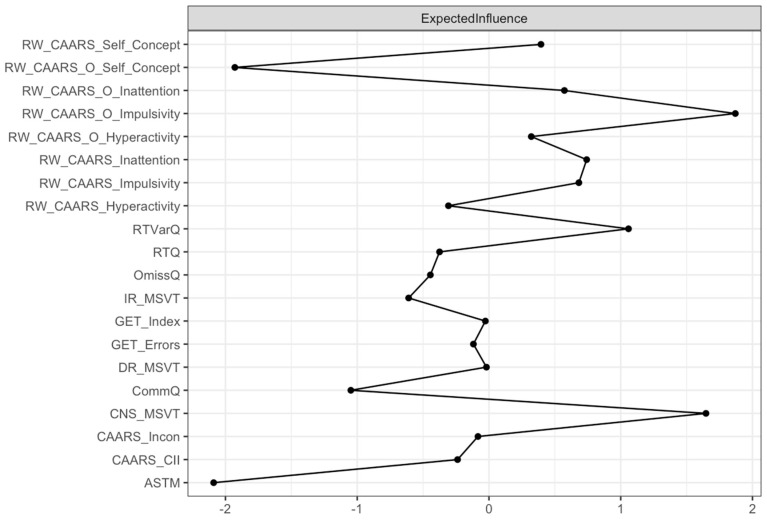
Node Expected Influence in the full sample (*N* = 461). Note: higher standardized Z-scores indicate greater expected influence, meaning that nodes with higher scores have stronger and closer connections with other neuropsychological test variables in the network: PVT (ASTM, MSVT, GET), QB+ (RTQ, RTVarQ, OmissQ, CommQ), and SVT (CAARS_CII, CAARS_Incon) variables, and the CAARS_S/O (Inattention, Hyperactivity, Impulsivity, Self-Concept) scales. ASTM = Amsterdam Short-Term Memory test; MSVT_IR = Medical Symptom Validity Test (Immediate Recall); MSVT_DR = Medical Symptom Validity Test (Delayed Recognition); MSVT_CNS = Medical Symptom Validity Test (Consistency Scores); GET_Index = Groningen Effort Test (Index); GET_Errors = Groningen Effort Test (Errors); RTQ = Reaction Time Q-score (Qb+); RTVarQ = Reaction Time Variation Q-score (Qb+); OmissQ = Omission Error Q-score (Qb+); CommQ = Commission Error Q-score (Qb+); RW_CAARS_Inattention = CAARS-L:S (Inattention); RW_CAARS_Hyperactivity = CAARS-L:S (Hyperactivity); RW_CAARS_Impulsivity = CAARS-L:S (Impulsivity); RW_CAARS_Self_Concept = CAARS-L:S (Self-Concept); RW_CAARS_O_Inattention = CAARS-L:O (Inattention); RW_CAARS_O_Hyperactivity = CAARS-L:O (Hyperactivity); RW_CAARS_Impulsivity = CAARS-L:O (Impulsivity); RW_CAARS_O_Self_Concept = CAARS-L:O (Self-Concept); CAARS_CII = CAARS-L:S (Infrequency Index); CAARS_Incon = CAARS-L:S (Inconsistency Index).

**Figure 4 behavsci-15-01108-f004:**
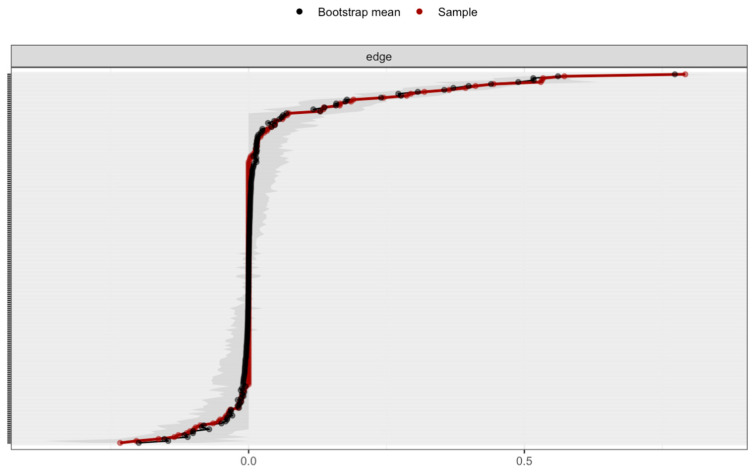
Edge weight accuracy estimation in the full sample. Note. Bootstrapped CIs of the estimated edge weights for the estimated network of the full sample. Each horizontal line represents an edge in the network, arranged from the highest to the lowest edge weight. The red line shows the sample edge weights, while the black line represents the bootstrap mean edge weights. The gray area indicates the bootstrapped confidence intervals, and the y-axis labels have been removed to reduce visual clutter.

**Table 1 behavsci-15-01108-t001:** Sample descriptives and clinical information.

	*N*	M_age_	SD_age_	Minimum_age_	Maximum_age_	ADHD Diagnosis (%)
Total sample	461	33.2	10.4	17	77	388 (84.2%)
Sex						
Females	268	32.9	10.5	20	77	230 (85.8%)
Males	193	33.6	10.3	17	60	158 (81.9%)
Comorbidity ^1^	158	34.6	11.05	18	77	128 (81.0%)
Highest educational level ^2^						
Unknown	3	37.2	16.0	21	52	3 (100%)
No school degree	1	-	-	-	-	1 (100%)
Basic school degree	79	34.9	13.3	17	77	67 (84.8%)
Secondary school degree	71	36.5	10.1	22	60	68 (95.8%)
Grammar school degree	178	29.5	8.62	20	64	147 (82.6%)
University degree	129	35.4	9.30	20	65	102 (79.1%)

Note. *N* = number of cases, M = mean, SD = standard deviation. ^1^ Comorbidity is determined using SCID (SCID-I; [106]) reported and classified according to ICD-10 mental disorders (F1–F9), excluding ADHD as a potential diagnosis. ^2^ Education = no school degree/no formal degree (less than 9 years of schooling); basic school degree (9–10 years of school); secondary school degree with additional vocational training (10–12 years of schooling); grammar school degree with university entrance qualification (usually 12–13 years of schooling); university degree (usually 16–17 years of schooling).

**Table 2 behavsci-15-01108-t002:** Measures of PVTs, SVTs, CPT, and ADHD symptom reports (*N* = 461).

						Below Average ^1^	
	M	SD	Minimum	Maximum	Positive Results (%)	Female (%)	Male (%)
GET							
Index	0.09	3.32	−17	15	141 (30.6%)		
Errors	7.11	6.88	0	57	59 (12.8%)		
Index + Errors					47 (10.2%)		
ASTM	86.9	5.03	56	90	70 (15.2%)		
MSVT					8 (1.7%)		
IR	99.4	2.36	80	100	1 (0.3%)		
DR	98.9	3.27	75	100	5 (1.1%)		
CNS	98.5	4.22	65	100	8 (1.7%)		
Qb+ ^2^							
RTQ	0.68	1.01	−2	4.9		38 (14.2%)	51 (26.4%)
RTVarQ	1.08	1.14	−1.8	4.5		91 (34.0%)	65 (33.7%)
OmissionQ	1.3	1.07	−1.5	3.5		144 (53.7%)	84 (43.5%)
CommissionQ	0.63	1.23	−1.6	4.7		75 (28.0%)	37 (19.2%)
CAARS—Inconsistency	5.94	2.18	1	13	107 (23.2%)		
CAARS Infrequency Index (CII)	17.7	5.59	1	35	109 (23.6%)		
CAARS-L:S ^3^							
Inattention	78.4	10.6	34	90		240 (89.5%)	173 (89.6%)
Hyperactivity	69.9	14.0	34	90		178 (66.4%)	124 (64.2%)
Impulsivity	70.1	12.3	36	90		204 (76.2%)	111 (57.5%)
Self-Concept Problems	72.0	12.1	35	90		196 (73.1%)	147 (76.2%)
CAARS-L:O ^4^							
Inattention	71.6	12.9	37	90		192 (71.6%)	128 (66.3%)
Hyperactivity	64.1	14.5	35	90		133 (49.6%)	87 (45.1%)
Impulsivity	65.2	12.7	35	90		147 (54.9%)	89 (46.1%)
Self-Concept Problems	67.6	12.2	36	90		169 (63.1%)	120 (62.2%)

Note. M = mean, SD = standard deviation. ^1^ Below average is defined as PR < 8 on neuropsychological tests (Qb+) and PR > 92 on questionnaire scores. ^2^ Qb+ = Q-based norm scores (adjusted for age and gender). ^3,4^ CAARS-L:S; CAARS-L:O = T-scores, adjusted for age and gender. ASTM = Amsterdam Short-Term Memory test; MSVT IR, DR, CNS = Medical Symptom Validity Test (Immediate Recall, Delayed Recognition, Consistency Scores); GET Index, Errors = Groningen Effort Test (Index, Errors); Qb+ = Quantified Behaviour Test; RTQ = Reaction Time Q-score; RTVarQ = Reaction Time Variation Q-score; OmissionQ = Omission Error Q-score; CommissionQ = Commission Error Q-score; CAARS—Inconsistency = CAARS-L:S (Inconsistency Index); CAARS Infrequency Index (CII) = CAARS-L:S (Infrequency Index); CAARS-L:S = Conners Adult ADHD Rating Scale, Longform Self-report (Inattention, Hyperactivity, Impulsivity, Self-Concept Problems); CAARS-L:O = Conners Adult ADHD Rating Scale, Longform Observer report (Inattention, Hyperactivity, Impulsivity, Self-Concept Problems).

**Table 3 behavsci-15-01108-t003:** Item allocation to factor structure (*N* = 461).

Measures	Conceptual Three-Factor Model	Conceptual Two-Factor Model	Conceptual One-Factor Model
GET			
Index	Performance Validity	Test Performance	One Factor
Errors	Performance Validity	Test Performance	One Factor
ASTM	Performance Validity	Test Performance	One Factor
MSVT			
IR	Performance Validity	Test Performance	One Factor
DR	Performance Validity	Test Performance	One Factor
CNS	Performance Validity	Test Performance	One Factor
Qb+			
RTQ	Continuous Performance	Test Performance	One Factor
RTVarQ	Continuous Performance	Test Performance	One Factor
OmissionQ	Continuous Performance	Test Performance	One Factor
CommissionQ	Continuous Performance	Test Performance	One Factor
CAARS—Inconsistency	Symptom Report	Symptom Report	One Factor
CAARS Infrequency Index (CII)	Symptom Report	Symptom Report	One Factor
CAARS-L:S			
Inattention	Symptom Report	Symptom Report	One Factor
Hyperactivity	Symptom Report	Symptom Report	One Factor
Impulsivity	Symptom Report	Symptom Report	One Factor
Self-Concept Problems	Symptom Report	Symptom Report	One Factor
CAARS-L:O			
Inattention	Symptom Report	Symptom Report	One Factor
Hyperactivity	Symptom Report	Symptom Report	One Factor
Impulsivity	Symptom Report	Symptom Report	One Factor
Self-Concept Problems	Symptom Report	Symptom Report	One Factor

Note. ASTM = Amsterdam Short-Term Memory test; MSVT IR, DR, CNS = Medical Symptom Validity Test (Immediate Recall, Delayed Recognition, Consistency Scores); GET Index, Errors = Groningen Effort Test (Index, Errors); Qb+ = Quantified Behaviour Test; RTQ = Reaction Time Q-score; RTVarQ = Reaction Time Variation Q-score; OmissionQ = Omission Error Q-score; CommissionQ = Commission Error Q-score; CAARS—Inconsistency = CAARS-L:S (Inconsistency Index); CAARS Infrequency Index (CII) = CAARS-L:S (Infrequency Index); CAARS-L:S = Conners Adult ADHD Rating Scale, Longform Self-report (Inattention, Hyperactivity, Impulsivity, Self-Concept Problems); CAARS-L:O = Conners Adult ADHD Rating Scale, Longform Observer report (Inattention, Hyperactivity, Impulsivity, Self-Concept Problems).

**Table 4 behavsci-15-01108-t004:** Measures of fit ^1^ for the one-, two-, and three-factor confirmatory factor analysis model.

Goodness of Fit	χ^2^	χ^2^/df	RMSEA	CFI	TLI
Three-factor Model	1550.4	9.28	0.136	0.669	0.623
Two-factor Model	1945.7	11.5	0.154	0.570	0.517
One-factor Model	2835.9	16.7	0.19	0.34	0.27

Note. RMSEA = root mean square error of approximation; TLI = Tucker–Lewis index; CFI = comparative fit index. Model–data fit is considered acceptable when χ^2^/df ranges from 2 to 5, RMSEA is below 0.06, and CFI and TLI estimates exceed 0.80. A fit is considered good when χ^2^/df is close to 1, and CFI and TLI estimates surpass 0.95 ([50]). ^1^ The measures of fit are represented as robust variables and z-standardized.

**Table 5 behavsci-15-01108-t005:** Multivariate Logistic Regression models and odds ratios based on descriptive and clinical information to predict the pass/fail PVT outcomes.

Predictors	ß	SE	Z	OR	*p*
**ASTM ^1^**					
Age	−0.002	0.013	−0.183	1.00	0.855
Sex (Male/Female)	0.002	0.305	0.006	1.00	0.995
ADHD (No/Yes)	0.402	0.431	0.933	1.49	0.351
Comorbidity ^2^ (No/Yes)	0.054	0.315	0.171	1.06	0.865
Cognitive Deficit ^3^ (No/Yes)	−1.23	0.374	−3.28	0.29	<0.001 ***
Education Level ^4^	0.302	0.132	2.30	1.35	<0.05 *
CAARS-L:S					
Inattention	0.211	0.513	0.412	1.24	0.680
Hyperactivity	0.222	0.351	0.632	1.25	0.528
Impulsivity	−0.791	0.402	−1.97	0.45	<0.05 *
CAARS-L:O					
Inattention	−0.140	0.364	−0.383	0.87	0.702
Hyperactivity	−0.499	0.350	−1.44	0.61	0.150
Impulsivity	0.987	0.360	2.78	2.68	<0.01 **
**MSVT ^1^**					
Age	−0.022	0.033	−0.661	0.98	0.508
Sex (Male/Female)	0.859	0.773	1.11	2.36	0.267
ADHD (No/Yes)	−17.4	2849.5	−0.006	0.00	0.995
Comorbidity ^2^ (No/Yes)	−0.812	0.755	−1.08	0.44	0.282
Cognitive Deficit ^3^ (No/Yes)	−17.8	2007.6	−0.009	0.00	0.993
Education Level ^4^	−0.155	0.343	−0.452	0.86	0.651
CAARS-L:S					
Inattention	−17.2	3516.7	−0.005	0.00	0.996
Hyperactivity	−0.605	0.909	−0.666	0.55	0.506
Impulsivity	1.35	0.873	1.55	3.87	0.121
CAARS-L:O					
Inattention	−0.855	1.15	−0.741	0.43	0.459
Hyperactivity	0.241	0.892	0.270	1.27	0.787
Impulsivity	0.101	0.952	0.106	1.11	0.916
**GET ^1^**					
Age	−0.022	0.010	−2.23	0.98	<0.05 *
Sex (Male/Female)	−0.286	0.213	−1.35	0.75	0.178
ADHD (No/Yes)	−0.026	0.311	−0.08	0.97	0.933
Comorbidity ^2^ (No/Yes)	−0.122	0.215	−0.564	0.89	0.573
Cognitive Deficit ^3^ (No/Yes)	−0.292	0.217	−1.35	0.75	0.178
Education Level ^4^	0.126	0.094	1.35	1.13	0.178
CAARS-L:S					
Inattention	−0.164	0.381	−0.431	0.85	0.667
Hyperactivity	−0.079	0.246	−0.324	0.92	0.746
Impulsivity	−0.275	0.263	−1.04	0.76	0.296
CAARS-L:O					
Inattention	−0.272	0.254	−1.07	0.76	0.286
Hyperactivity	−0.126	0.237	−0.531	0.88	0.595
Impulsivity	0.094	0.240	0.393	1.10	0.694

Note. ^1^ PVT measures (ASTM, MSVT, GET) depicted as pass/fail binary variable according to the cut-off criteria described in the test manuals (see; [35]; [40]; [88]); ASTM = Amsterdam Short-Term Memory test; MSVT = Medical Symptom Validity Test; GET = Groningen Effort Test. ^2^ Comorbidity is determined using SCID-I ([106]) reported and classified according to ICD-10 mental disorders (F1–F9), excluding ADHD as a potential diagnosis. Depicted as a binary variable indicating the presence/absence of one or more comorbidities. ^3^ Cognitive Deficit derived from the Qb+ measures (RTQ, RTVarQ, OmissionQ, CommissionQ), any of the four measures Q-score > 1.5; pass/fail binary variable. ^4^ Data missing from 3 participants, due to unknown education level. Depicted as a continuous variable with the following encoding based on years of schooling: 1 represents no school degree or no formal degree (less than 9 years of schooling), 2 represents a basic school degree (9–10 years of schooling), 3 represents a secondary school degree with additional vocational training (10–12 years of schooling), 4 represents a grammar school degree with university entrance qualification (usually 12–13 years of schooling), and 5 represents a university degree (usually 16–17 years of schooling). Statistically significant at *p* < 0.05 *, *p* < 0.01 **, *p* < 0.001 ***. CAARS-L:S = Conners Adult ADHD Rating Scale, Longform Self-report (Inattention, Hyperactivity, Impulsivity, Self-Concept Problems); CAARS-L:O = Conners Adult ADHD Rating Scale, Longform Observer report (Inattention, Hyperactivity, Impulsivity, Self-Concept Problems).

## Data Availability

The data presented in this study are available on request from the corresponding author.

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
