# Peer review of "Joint Factor Performance Validity?—Network and Factor Structure of Performance Validity Measures in the Clinical Evaluation of Adult ADHD"

_behavsci, 2025, doi:10.3390/bs15081108_

Round 1

Reviewer 1 Report

Comments and Suggestions for Authors

This is a well-written manuscript reporting on a well-designed study with high heuristic value. The data analysis is thorough and cutting edge. The topic is timely and of great clinical relevance. My only suggestion is to carefully proofread for typos and the correct (APA style) formatting of the references.

Author Response

Reviewer 1

We appreciate Reviewer 1’s thorough reading of the manuscript and their positive evaluation of the study’s methodology, relevance and contribution to the literature. Below, we address each point in the review:

Comment 1: “… For example, they provide valuable evidence to refute recent criticisms by Leonhard that “all PVTs are collinear”. The authors may consider making this explicit in their Discussion”

Response 1: This point has been addressed in the revised Discussion section (p. 9, in 698-702). We now explicitly articulate how our findings contradict the assumption that “all PVTs are collinear”. This argument not only strengthens the viewpoint suggested by Rosenfeld (2000), Boone (2009) and Erdodi’s (2019) domain-specific hypothesis but also the theoretical implications of our findings.

Comment 2:My main suggestion is to carefully proofread for typos and the correct (APA style) formatting of the references” 

Response 2: The manuscript has been carefully proofread to eliminate typographical errors. We also reviewed all references and in-text citations to ensure consistency with APA 7th guidelines.

Reviewer 2 Report

Comments and Suggestions for Authors

This manuscript reports results from a set of clinically referred adult patients in order to evaluate potential ADHD (N = 461). This study evaluated underlying relationships and constructs of PVTs, SVTs, reported symptoms, and cognitive performance. In contrast to some prior literature, the authors did not find the hypothesized three construct result of PVT, cognitive tests, SVTs/symptoms, but rather a 4 factor solution was best, which seemed to hand on to intra-test scores for the most part. Additionally, the ASTM was related to various demographic factors, but this was not the case for the MSVT nor the GET. The authors indicate that results suggest a multi-modal approach to evaluating response bias, particularly with PVTs, suggesting a domain specific relationship as opposed to a single, underlying PVT construct. In general, the paper is well written and the topic appropriate and grounded in current response bias research, while also connecting to ADHD research more generally, and increasingly popular and clinically-relevant topic. Some of the statistical analyses are beyond my own scope, thus a stats review might be considered; however, I have only a few minor suggestions to make.

5 ln205-212: The test is described as memorizing 5 words, but then after the distractor recalling 3 words. Is it 5 or 3? Readers outside Europe might be less familiar with this test (as I am), so clarification would be helpful.

6 lns 219-227: Please state whether or not the Genuine Memory Impairment profile was or was not used. Here it makes sense to NOT consider it given the nature of ADHD, but a sentence or two needs added just to clarify and support the decision.

6 ln 236: Here again, the GET is a less common test, at least in the US. Please describe the “Index” score.

9 ln 657: Would recommend changing the term “malingering factor” as malingering is not measured here, rather response bias more broadly is.

Author Response

Reviewer 2

We appreciate Reviewer’s 2 careful review and their recognition of the manuscript’s relevance to both ADHD and response bias research. Below, we address each point in the review:

Comment 1:5 ln205-212: The test is described as memorizing 5 words, but then after the distractor recalling 3 words. Is it 5 or 3? Readers outside Europe might be less familiar with this test (as I am), so clarification would be helpful

Response 1: We have revised the manuscript to clarify that the ASTM requires participants to memorize five words initially, but that only three of those are assessed in the critical recall trials following the distractor task (arithmetic problem). The text now includes a clearer explanation to assist readers unfamiliar with the ASTM (p. 6, in 225-229).

Comment 2: “Please state whether or not the Genuine Memory Impairment profile was or was not used. Here it makes sense to NOT consider it given the nature of ADHD, but a sentence or two needs added just to clarify and support the decision”

Response 2: We have added clarification that the Genuine Memory Impairment Profile was not used in the present study. This decision was based on clinical considerations specific to ADHD, as the profile is cautioned against for individuals without significant functional decline, which was not expected or assessed in our sample (p. 6, in 241-245).

Comment 3:Here again, the GET is a less common test, at least in the US. Please describe the “Index” score”

Response 3: A brief description of the GET Index score has been added to the GET section. Specifically, we clarify that the GET Index is calculated by combining processing speed and accuracy measure within each block of trials (p. 6, in 257-260).

Comment 4: “Would recommend changing the term “malingering factor” as malingering is not measured here, rather response bias more broadly is”

Response 4: We agree that the term “malingering” is too specific given the scope of the used measures. The manuscript did not refer to a “malingering factor” but instead mentioned “malingering patterns”, which we have now revised in the manuscript to “response bias patterns”. The terminology more accurately reflects the constructs measured and avoids implying intentional deception, which was not directly assessed in the present study (p. 9, in 692-695).